# Zap Q-Learning

**Adithya M. Devraj**      **Sean P. Meyn**
Department of Electrical and Computer Engineering,
University of Florida,
Gainesville, FL 32608.
adithyamdevraj@ufl.edu, meyn@ece.ufl.edu

## Abstract

The Zap Q-learning algorithm introduced in this paper is an improvement of Watkins' original algorithm and recent competitors in several respects. It is a matrix-gain algorithm designed so that its asymptotic variance is optimal. Moreover, an ODE analysis suggests that the transient behavior is a close match to a deterministic Newton-Raphson implementation. This is made possible by a two time-scale update equation for the matrix gain sequence. The analysis suggests that the approach will lead to stable and efficient computation even for non-ideal parameterized settings. Numerical experiments confirm the quick convergence, even in such non-ideal cases.

## 1   Introduction

It is recognized that algorithms for reinforcement learning such as TD- and Q-learning can be slow to converge. The poor performance of Watkins' Q-learning algorithm was first quantified in [25], and since then many papers have appeared with proposed improvements, such as [9, 1].

An emphasis in much of the literature is computation of finite-time PAC (probably almost correct) bounds as a metric for performance. Explicit bounds were obtained in [25] for Watkins' algorithm, and in [1] for the "speedy" Q-learning algorithm that was introduced by these authors. A general theory is presented in [18] for stochastic approximation algorithms.

In each of the models considered in prior work, the update equation for the parameter estimates can be expressed

$$\theta_{n+1} = \theta_n + \alpha_n[\overline{f}(\theta_n) + \Delta_{n+1}], \quad n \geq 0, \tag{1}$$

in which $\{\alpha_n\}$ is a positive gain sequence, and $\{\Delta_n\}$ is a martingale difference sequence. This representation is critical in analysis, but unfortunately is not typical in reinforcement learning applications outside of these versions of Q-learning. For Markovian models, the usual transformation used to obtain a representation similar to (1) results in an error sequence $\{\Delta_n\}$ that is the sum of a martingale difference sequence and a *telescoping sequence* [15]. It is the telescoping sequence that prevents easy analysis of Markovian models.

This gap in the research literature carries over to the general theory of Markov chains. Examples of concentration bounds for i.i.d. sequences or martingale-difference sequences include the finite-time bounds of Hoeffding and Bennett. Extensions to Markovian models either offer very crude bounds [17], or restrictive assumptions [14, 11]; this remains an active area of research [20].

In contrast, *asymptotic* theory for stochastic approximation (as well as general state space Markov chains) is mature. Large Deviations or Central Limit Theorem (CLT) limits hold under very general assumptions [3, 13, 4]. The CLT will be a guide to algorithm design in the present paper. For a typical stochastic approximation algorithm, this takes the following form: denoting $\{\widetilde{\theta}_n := \theta_n - \theta^* : n \geq 0\}$ to be the error sequence, under general conditions the scaled sequence $\{\sqrt{n}\widetilde{\theta}_n : n \geq 1\}$

converges in distribution to a Gaussian distribution, $\mathcal{N}(0, \Sigma_\theta)$. Typically, the scaled covariance is also convergent to the limit, which is known as the *asymptotic covariance*:

$$\Sigma_\theta = \lim_{n \to \infty} n \mathsf{E}[\widetilde{\theta}_n \widetilde{\theta}_n^T]. \tag{2}$$

An asymptotic bound such as (2) may not be satisfying for practitioners of stochastic optimization or reinforcement learning, given the success of finite-$n$ performance bounds in prior research. However, the fact that the asymptotic covariance $\Sigma_\theta$ has a simple representation, and can therefore be easily improved or optimized, makes it a compelling tool to consider. Moreover, as the examples in this paper suggest, the asymptotic covariance is often a good predictor of finite-time performance, since the CLT approximation is accurate for reasonable values of $n$.

Two approaches are known for optimizing the asymptotic covariance. First is the remarkable averaging technique introduced in [21, 22, 24] (also see [12]). Second is *Stochastic Newton-Raphson*, based on a special choice of matrix gain for the algorithm [13, 23]. The algorithms proposed here use the second approach.

Matrix gain variants of TD-learning [10, 19, 29, 30] and Q-learning [27] are available in the literature, but none are based on optimizing the asymptotic variance. It is a fortunate coincidence that LSTD($\lambda$) of [6] achieves this goal [8].

In addition to accelerating the convergence rate of the standard Q-learning algorithm, it is hoped that this paper will lead to entirely new algorithms. In particular, there is little theory to support Q-learning in non-ideal settings in which the optimal "$Q$-function" does not lie in the parameterized function class. Convergence results have been obtained for a class of optimal stopping problems [31], and for deterministic models [16]. There is now intense practical interest, despite an incomplete theory. A stronger supporting theory will surely lead to more efficient algorithms.

**Contributions**   A new class of Q-learning algorithms is proposed, called *Zap Q-learning*, designed to more accurately mimic the classical Newton-Raphson algorithm. It is based on a two time-scale stochastic approximation algorithm, constructed so that the matrix gain tracks the gain that would be used in a deterministic Newton-Raphson method.
A full analysis is presented for the special case of a complete parameterization (similar to the setting of Watkins' algorithm [28]). It is found that the associated ODE has a remarkable and simple representation, which implies consistency under suitable assumptions. Extensions to non-ideal parameterized settings are also proposed, and numerical experiments show dramatic variance reductions. Moreover, results obtained from finite-$n$ experiments show close solidarity with asymptotic theory.

The remainder of the paper is organized as follows. The new Zap Q-learning algorithm is introduced in Section 2, which contains a summary of the theory from extended version of this paper [8]. Numerical results are surveyed in Section 3, and conclusions are contained in Section 4.

## 2   Zap Q-Learning

Consider an MDP model with state space $\mathsf{X}$, action space $\mathsf{U}$, cost function $c \colon \mathsf{X} \times \mathsf{U} \to \mathbb{R}$, and discount factor $\beta \in (0, 1)$. It is assumed that the state and action space are finite: denote $\ell = |\mathsf{X}|$, $\ell_u = |\mathsf{U}|$, and $P_u$ the $\ell \times \ell$ conditional transition probability matrix, conditioned on $u \in \mathsf{U}$. The state-action process $(\boldsymbol{X}, \boldsymbol{U})$ is adapted to a filtration $\{\mathcal{F}_n : n \geq 0\}$, and Q1 is assumed throughout:
**Q1:** The joint process $(\boldsymbol{X}, \boldsymbol{U})$ is an irreducible Markov chain, with unique invariant pmf $\varpi$.

The minimal value function is the unique solution to the discounted-cost optimality equation:

$$h^*(x) = \min_{u \in \mathsf{U}} Q^*(x, u) := \min_{u \in \mathsf{U}} \{ c(x, u) + \beta \sum_{x' \in \mathsf{X}} P_u(x, x') h^*(x') \}, \quad x \in \mathsf{X}.$$

The "Q-function" solves a similar fixed point equation:

$$Q^*(x, u) = c(x, u) + \beta \sum_{x' \in \mathsf{X}} P_u(x, x') \underline{Q}^*(x'), \quad x \in \mathsf{X}, \ u \in \mathsf{U}, \tag{3}$$

in which $\underline{Q}(x) := \min_{u \in \mathsf{U}} Q(x, u)$ for any function $Q \colon \mathsf{X} \times \mathsf{U} \to \mathbb{R}$.

Given any function $\varsigma \colon \mathsf{X} \times \mathsf{U} \to \mathbb{R}$, let $\mathcal{Q}(\varsigma)$ denote the corresponding solution to the fixed point equation (3), with $c$ replaced by $\varsigma$: The function $q = \mathcal{Q}(\varsigma)$ is the solution to the fixed point equation,

$$q(x, u) = \varsigma(x, u) + \beta \sum_{x'} P_u(x, x') \min_{u'} q(x', u'), \quad x \in \mathsf{X}, \ u \in \mathsf{U}.$$

The mapping $\mathcal{Q}$ is a bijection on the set of real-valued functions on $\mathsf{X} \times \mathsf{U}$. It is also piecewise linear, concave and monotone (See [8] for proofs and discussions).

It is known that Watkins' Q-learning algorithm can be regarded as a stochastic approximation method [26, 5] to obtain the solution $\theta^* \in \mathbb{R}^d$ to the steady-state mean equations,

$$\mathsf{E}\big[\{c(X_n, U_n) + \beta \underline{Q}^{\theta^*}(X_{n+1}) - Q^{\theta^*}(X_n, U_n)\}\zeta_n(i)\big] = 0, \quad 1 \le i \le d \tag{4}$$

where $\{\zeta_n\}$ are $d$-dimensional $\mathcal{F}_n$-measurable functions and $Q^\theta = \theta^T \psi$ for basis functions $\{\psi_i : 1 \le i \le d\}$. In Watkins' algorithm $\zeta_n = \psi(X_n, U_n)$, and the basis functions are indicator functions: $\psi_k(x, u) = \mathbb{I}\{x = x^k, u = u^k\}$, $1 \le k \le d$, with $d = \ell \times \ell_u$ the total number of state-action pairs [26]. In this special case we identify $Q^{\theta^*} = Q^*$, and the parameter $\theta$ is identified with the estimate $Q^\theta$. A stochastic approximation algorithm to solve (4) coincides with Watkins' algorithm [28]:

$$\theta_{n+1} = \theta_n + \alpha_{n+1}\{c(X_n, U_n) + \beta\underline{\theta}_n(X_{n+1}) - \theta_n(X_n, U_n)\}\psi(X_n, U_n) \tag{5}$$

One very general technique that is used to analyze convergence of stochastic approximation algorithms is to consider the *associated limiting ODE*, which is the continuous-time, deterministic approximation of the original recursion [4, 5]. For (5), denoting the continuous time approximation of $\{\theta_n\}$ to be $\{q_t\}$, and under standard assumptions on the gain sequence $\{\alpha_n\}$, the associated ODE is of the form

$$\tfrac{d}{dt}q_t(x, u) = \varpi(x, u)\Big\{c(x, u) + \beta\sum_{x'} P_u(x, x')\min_{u'} q(x', u') - q_t(x, u)\Big\}. \tag{6}$$

Under Q1, $\{q_t\}$ converges to $Q^*$: A key step in the proof of convergence of $\{\theta_n\}$ to the same limit.

While Watkins' Q-learning (5) is consistent, it is argued in [8] that the asymptotic covariance of this algorithm is typically infinite. This conclusion is complementary to the finite-$n$ analysis of [25]:

**Theorem 2.1.** *Watkins' Q-learning algorithm with step-size $\alpha_n \equiv 1/n$ is consistent under Assumption Q1. Suppose that in addition $\max\limits_{x,u} \varpi(x, u) \le \frac{1}{2}(1 - \beta)^{-1}$, and the conditional variance of $h^*(X_t)$ is positive:*

$$\sum_{x,x',u} \varpi(x, u)P_u(x, x')[h^*(x') - P_u h^*(x)]^2 > 0$$

*Then the asymptotic covariance is infinite:* $\lim\limits_{n \to \infty} n\mathsf{E}[\|\theta_n - \theta^*\|^2] = \infty.$ $\qquad\square$

The assumption $\max_{x,u} \varpi(x, u) \le \frac{1}{2}(1 - \beta)^{-1}$ is satisfied whenever $\beta \ge \frac{1}{2}$.

Matrix-gain stochastic approximation algorithms have appeared in previous literature. In particular, matrix gain techniques have been used to speed-up the rate of convergence of Q-learning (see [7] and the second example in Section 3). The general *G-Q($\lambda$)* algorithm is described as follows, based on a sequence of $d \times d$ matrices $\boldsymbol{G} = \{G_n\}$ and $\lambda \in [0, 1]$: For initialization $\theta_0, \zeta_0 \in \mathbb{R}^d$, the sequence of estimates are defined recursively:

$$\begin{aligned}
\theta_{n+1} &= \theta_n + \alpha_{n+1}G_{n+1}\zeta_n d_{n+1} \\
d_{n+1} &= c(X_n, U_n) + \beta\underline{Q}^{\theta_n}(X_{n+1}) - Q^{\theta_n}(X_n, U_n) \\
\zeta_{n+1} &= \lambda\beta\zeta_n + \psi(X_{n+1}, U_{n+1})
\end{aligned} \tag{7}$$

The special case based on stochastic Newton-Raphson is *Zap Q($\lambda$)-learning:*

---

**Algorithm 1** Zap Q($\lambda$)-learning

---

**Input:** $\theta_0 \in \mathbb{R}^d$, $\zeta_0 = \psi(X_0, U_0)$, $\widehat{A}_0 \in \mathbb{R}^{d \times d}$, $n = 0$, $T \in \mathcal{Z}^+$ $\qquad\qquad\qquad\qquad \triangleright$ Initialization
1: **repeat**
2: $\qquad \phi_n(X_{n+1}) := \arg\min_u Q^{\theta_n}(X_{n+1}, u);$
3: $\qquad d_{n+1} := c(X_n, U_n) + \beta Q^{\theta_n}(X_{n+1}, \phi_n(X_{n+1})) - Q^{\theta_n}(X_n, U_n);$ $\quad \triangleright$ Temporal difference
4: $\qquad A_{n+1} := \zeta_n\big[\beta\psi(X_{n+1}, \phi_n(X_{n+1})) - \psi(X_n, U_n)\big]^T;$
5: $\qquad \widehat{A}_{n+1} = \widehat{A}_n + \gamma_{n+1}\big[A_{n+1} - \widehat{A}_n\big];$ $\qquad\qquad\qquad\qquad\qquad \triangleright$ Matrix gain update rule
6: $\qquad \theta_{n+1} = \theta_n - \alpha_{n+1}\widehat{A}_{n+1}^{-1}\zeta_n d_{n+1};$ $\qquad\qquad\qquad\qquad\qquad\qquad \triangleright$ Zap-Q update rule
7: $\qquad \zeta_{n+1} := \lambda\beta\zeta_n + \psi(X_{n+1}, U_{n+1});$ $\qquad\qquad\qquad\qquad \triangleright$ Eligibility vector update rule
8: $\qquad n = n + 1$
9: **until** $n \ge T$

---

A special case is considered in the analysis here: the basis is chosen as in Watkins' algorithm, $\lambda = 0$, and $\alpha_n \equiv 1/n$. An equivalent representation for the parameter recursion is thus

$$\theta_{n+1} = \theta_n - \alpha_{n+1}\widehat{A}_{n+1}^{-1}\{\Psi_n c + A_{n+1}\theta_n\},$$

in which $c$ and $\theta_n$ are treated as $d$-dimensional vectors rather than functions on $\mathsf{X} \times \mathsf{U}$, and $\Psi_n = \psi(X_n, U_n)\psi(X_n, U_n)^T$.

Part of the analysis is based on a recursion for the following $d$-dimensional sequence:

$$\widehat{C}_n = -\Pi^{-1}\widehat{A}_n\theta_n, \quad n \geq 1,$$

where $\Pi$ is the $d \times d$ diagonal matrix with entries $\varpi$ (the steady-state distribution of $(\boldsymbol{X}, \boldsymbol{U})$). The sequence $\{\widehat{C}_n\}$ admits a very simple recursion in the special case $\boldsymbol{\gamma} \equiv \boldsymbol{\alpha}$:

$$\widehat{C}_{n+1} = \widehat{C}_n + \alpha_{n+1}[\Pi^{-1}\Psi_n c - \widehat{C}_n]. \tag{8}$$

It follows that $\widehat{C}_n$ converges to $c$ as $n \to \infty$, since (8) is essentially a Monte-Carlo average of $\{\Pi^{-1}\Psi_n c : n \geq 0\}$. Analysis for this case is complicated since $\widehat{A}_n$ is obtained as a uniform average of $\{A_n\}$.

The main contributions of this paper concern a two time-scale implementation for which

$$\sum \gamma_n = \infty \quad \sum \gamma_n^2 < \infty \quad and \quad \lim_{n \to \infty} \frac{\alpha_n}{\gamma_n} = 0. \tag{9}$$

In our analysis, we restrict to $\gamma_n \equiv 1/n^\rho$, for some fixed $\rho \in (\frac{1}{2}, 1)$. Through ODE analysis, it is argued that the Zap Q-learning algorithm closely resembles an implementation of Newton-Raphson in this case. This analysis suggests that $\{\widehat{A}_n\}$ more closely tracks the mean of $\{A_n\}$. Theorem 2.2 summarizes the main results under Q1, and the following additional assumptions:

**Q2:** The optimal policy $\phi^*$ is unique.

**Q3:** The sequence of policies $\{\phi_n\}$ satisfy $\displaystyle\sum_{n=1}^{\infty} \gamma_n \mathbb{I}\{\phi_{n+1} \neq \phi_n\} < \infty, \quad a.s..$

The assumption Q3 is used to address the discontinuity in the recursion for $\{\widehat{A}_n\}$ resulting from the dependence of $A_{n+1}$ on $\phi_n$.

**Theorem 2.2.** *Suppose that Assumptions Q1–Q3 hold, and the gain sequences $\boldsymbol{\alpha}$ and $\boldsymbol{\gamma}$ satisfy:*

$$\alpha_n = n^{-1}, \quad \gamma_n = n^{-\rho}, \quad n \geq 1,$$

*for some fixed $\rho \in (\frac{1}{2}, 1)$. Then,*

(i) *The parameter sequence $\{\theta_n\}$ obtained using the Zap-Q algorithm converges to $Q^*$ a.s..*

(ii) *The asymptotic covariance (2) is minimized over all $\boldsymbol{G}$-Q(0) matrix gain versions of Watkins' Q-learning algorithm.*

(iii) *An ODE approximation holds for the sequence $\{\theta_n, \widehat{C}_n\}$, by continuous functions $(\boldsymbol{q}, \varsigma)$ satisfying*

$$q_t = \mathcal{Q}(\varsigma_t), \quad \frac{d}{dt}\varsigma_t = -\varsigma_t + c \tag{10}$$

*This ODE approximation is exponentially asymptotically stable, with $\displaystyle\lim_{t \to \infty} q_t = Q^*$.* $\square$

The ODE result (10) is an important aspect of this work. It says that the sequence $\{q_t\}$, a continuous time approximation of the parameter estimates $\{\theta_n\}$ that are obtained using the Zap Q-learning algorithm, evolves as the Q-function of some time-varying cost function $\varsigma_t$. Furthermore, this time-varying cost function $\varsigma_t$ has dynamics independent of $q_t$, and converges to $c$; the cost function defined in the MDP model. Convergence follows from the continuity of the mapping $\mathcal{Q}$:

$$\lim_{n \to \infty} \theta_n = \lim_{t \to \infty} q_t = \lim_{t \to \infty} \mathcal{Q}(\varsigma_t) = \mathcal{Q}(c) = Q^*.$$

The reader is referred to [8] for complete proofs and technical details.

## 3 Numerical Results

Results from numerical experiments are surveyed here to illustrate the performance of the Zap Q-learning algorithm.

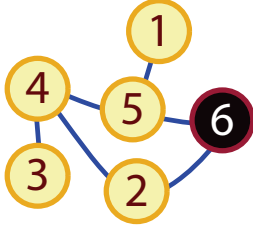

Figure 1: Graph for MDP

**Finite state-action MDP** Consider first a simple path-finding problem. The state space $\mathsf{X} = \{1, \ldots, 6\}$ coincides with the six nodes on the undirected graph shown in Fig. 1. The action space $\mathsf{U} = \{e_{x,x'}\}, x, x' \in \mathsf{X}$, consists of all feasible edges along which the agent can travel, including each "self-loop", $u = e_{x,x}$. The goal is to reach the state $x^* = 6$ and maximize the time spent there. The reader is referred to [8] for details on the cost function and other modeling assumptions.

Six variants of Q-learning were tested: Watkins' algorithm (5), Watkins' algorithm with Ruppert-Polyak-Juditsky (RPJ) averaging [21, 22, 24], Watkins' algorithm with a "polynomial learning rate" $\alpha_n \equiv n^{-0.6}$ [9], Speedy Q-learning [1], and two versions of Zap Q-learning: $\gamma_n \equiv \alpha_n \equiv n^{-1}$, and $\gamma_n \equiv \alpha_n^{0.85} \equiv n^{-0.85}$.

Fig. 2 shows the normalized trace of the asymptotic covariance of Watkins' algorithm with step-size $\alpha_n = g/n$, as a function of $g > 0$. Based on this observation or on Theorem 2.1, it follows that the asymptotic covariance is not finite for the standard Watkins' algorithm with $\alpha_n \equiv 1/n$. In simulations it was found that the parameter estimates are not close to $\theta^*$ even after many millions of iterations.

It was also found that Watkins' algorithm performed poorly in practice for any scalar gain. For example, more than half of the $10^3$ experiments using $\beta = 0.8$ and $g = 70$ resulted in values of $\theta_n(15)$ exceeding $\theta^*(15)$ by $10^4$ (with $\theta^*(15) \approx 500$), even with $n = 10^6$. The algorithm performed well with the introduction of projection (to ensure that the parameter estimates evolve on a bounded set) in the case $\beta = 0.8$. With $\beta = 0.99$, the performance was unacceptable for any scalar gain, even with projection.

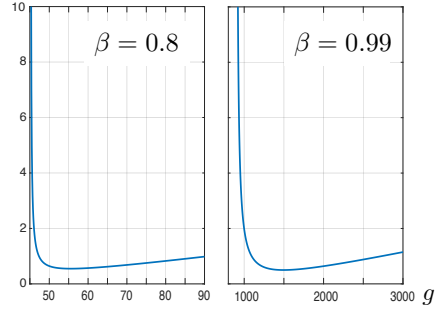

Figure 2: Normalized trace of the asymptotic covariance

Fig. 3 shows normalized histograms of $\{W_n^i(k) = \sqrt{n}(\theta_n^i(k) - \overline{\theta}_n(k)) : 1 \leq i \leq N\}$ for the projected Watkins Q-learning with gain $g = 70$, and the Zap algorithm, $\gamma_n \equiv \alpha_n^{0.85}$. The theoretical predictions were based on the solution to a Lyapunov equation [8]. Results for $\beta = 0.99$ contained in [8] show similar solidarity with asymptotic theory.

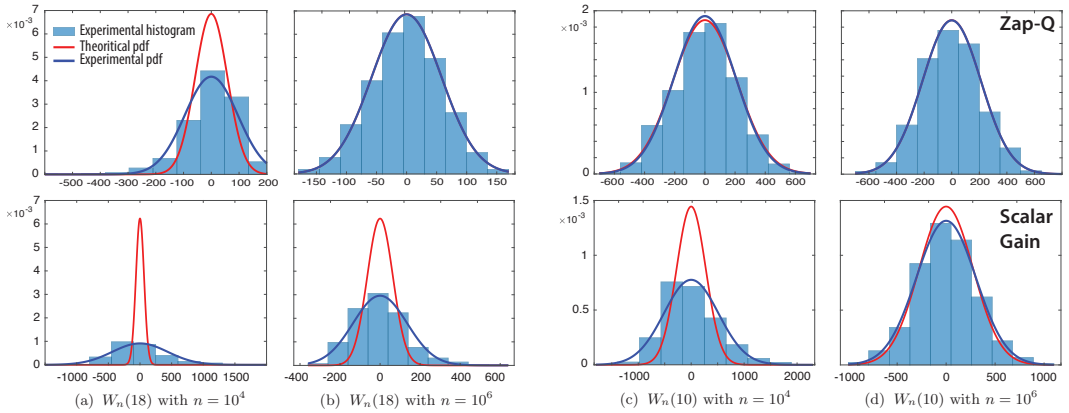

(a) $W_n(18)$ with $n = 10^4$     (b) $W_n(18)$ with $n = 10^6$     (c) $W_n(10)$ with $n = 10^4$     (d) $W_n(10)$ with $n = 10^6$

Figure 3: Asymptotic variance for Watkins' $g = 70$ and Zap Q-learning, $\gamma_n \equiv \alpha_n^{0.85}$; $\beta = 0.8$

***Bellman Error*** The Bellman error at iteration $n$ is denoted:
$$\mathcal{B}_n(x, u) = \theta_n(x, u) - r(x, u) - \beta \sum_{x' \in \mathsf{X}} P_u(x, x') \max_{u' \in \mathsf{U}} \theta_n(x', u').$$

This is identically zero if and only if $\theta_n = Q^*$. Fig. 4 contains plots of the maximal error $\overline{\mathcal{B}}_n = \max_{x,u} |\mathcal{B}_n(x, u)|$ for the six algorithms.

Though all six algorithms perform reasonably well when $\beta = 0.8$, Zap Q-learning is the only one that achieves near zero Bellman error within $n = 10^6$ iterations in the case $\beta = 0.99$. Moreover, the

performance of the two time-scale algorithm is clearly superior to the one time-scale algorithm. It is also observed that the Watkins algorithm with an optimized scalar gain (i.e., step-size $\alpha_n \equiv g^*/n$ with $g^*$ chosen so that the asymptotic variance is minimized) has the best performance among scalar-gain algorithms.

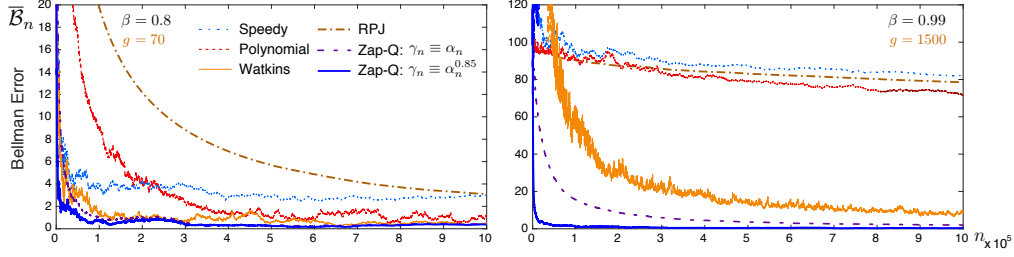

Figure 4: Maximum Bellman error $\{\overline{\mathcal{B}}_n : n \geq 0\}$ for the six Q-learning algorithms

Fig. 4 shows only the typical behavior — repeated trials were run to investigate the range of possible outcomes. Plots of the mean and $2\sigma$ confidence intervals of $\overline{\mathcal{B}}_n$ are shown in Fig. 5 for $\beta = 0.99$.

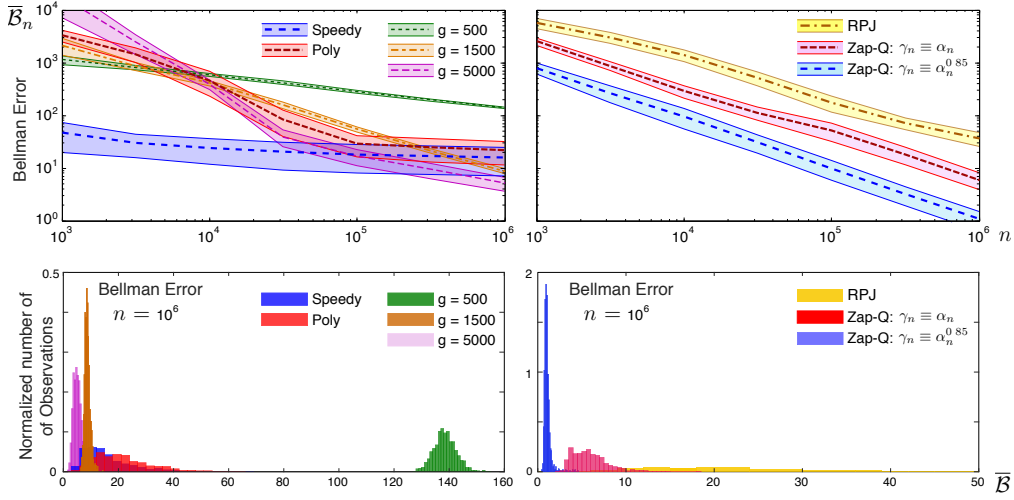

Figure 5: Simulation-based $2\sigma$ confidence intervals for the six Q-learning algorithms for the case $\beta = 0.99$

**Finance model**   The next example is taken from [27, 7]. The reader is referred to these references for complete details of the problem set-up and the reinforcement learning architecture used in this prior work. The example is of interest because it shows how the Zap Q-learning algorithm can be used with a more general basis, and also how the technique can be extended to optimal stopping time problems.

The Markovian state process for the model evolves in $\mathsf{X} = \mathbb{R}^{100}$. The "time to exercise" is modeled as a discrete valued stopping time $\tau$. The associated expected reward is defined as $\mathsf{E}[\beta^\tau r(X_\tau)]$, where $\beta \in (0,1)$, $r(X_n) := X_n(100) = \widetilde{p}_n/\widetilde{p}_{n-100}$, and $\{\widetilde{p}_t : t \in \mathbb{R}\}$ is a geometric Brownian motion (derived from an exogenous price-process). The objective of finding a policy that maximizes the expected reward is modeled as an optimal stopping time problem.

The value function is defined to be the supremum over all stopping times:
$$h^*(x) = \sup_{\tau > 0} \mathsf{E}[\beta^\tau r(X_\tau) \mid X_0 = x].$$
This solves the Bellman equation: For each $x \in \mathsf{X}$,
$$h^*(x) = \max\big(r(x), \beta\mathsf{E}[h^*(X_{n+1}) \mid X_n = x]\big).$$

The associated Q-function is denoted $Q^*(x) := \beta\mathsf{E}[h^*(X_{n+1}) \mid X_n = x]$, and solves a similar fixed point equation:
$$Q^*(x) = \beta\mathsf{E}[\max(r(X_{n+1}), Q^*(X_{n+1})) \mid X_n = x].$$

The Q(0)-learning algorithm considered in [27] is defined as follows:

$$\theta_{n+1} = \theta_n + \alpha_{n+1}\psi(X_n)\Big[\beta \max\big(X_{n+1}(100), Q^{\theta_n}(X_{n+1})\big) - Q^{\theta_n}(X_n)\Big], \quad n \geq 0.$$

In [7] the authors attempt to improve the performance of the Q(0) algorithm through the use of a sequence of matrix gains, which can be regarded as an instance of the $G$-Q(0)-learning algorithm defined in (7). For details see this prior work as well as the extended version of this paper [8].

A gain sequence $\{G_n\}$ was introduced in [7] to improve performance. Denoting $G$ and $A$ to be the steady state means of $\{G_n\}$ and $\{A_n\}$ respectively, the eigenvalues corresponding to the matrix $GA$ are shown on the right hand side of Fig. 6. It is observed that the sufficient condition for a finite asymptotic covariance are "just" satisfied in this algorithm: the maximum eigenvalue of $GA$ is approximately $\lambda \approx -0.525 < -\frac{1}{2}$ (see Theorem 2.1 of [8]). It is worth stressing that the finite asymptotic covariance was not a design goal in this prior work. It is only now on revisiting this paper that we find that the sufficient condition $\lambda < -\frac{1}{2}$ is satisfied.

The Zap Q-learning algorithm for this example is defined by the following recursion:

$$\theta_{n+1} = \theta_n - \alpha_{n+1}\widehat{A}_{n+1}^{-1}\psi(X_n)\Big[\beta \max\big(X_{n+1}(100), Q^{\theta_n}(X_{n+1})\big) - Q^{\theta_n}(X_n)\Big],$$

$$\widehat{A}_{n+1} = \widehat{A}_n + \gamma_n[A_{n+1} - \widehat{A}_n], \qquad A_{n+1} = \psi(X_n)\varphi^T(\theta_n, X_{n+1}),$$

$$\varphi(\theta_n, X_{n+1}) = \beta\psi(X_{n+1})\mathbb{I}\{Q^{\theta_n}(X_{n+1}) \geq X_{n+1}(100)\} - \psi(X_n).$$

***High performance despite ill-conditioned matrix gain***     The real part of the eigenvalues of $A$ are shown on a logarithmic scale on the left-hand side of Fig. 6. These eigenvalues have a wide spread: the ratio of the largest to the smallest real parts of the eigenvalues is of the order $10^4$. This presents a challenge in applying any method. In particular, it was found that the performance of any scalar-gain algorithm was extremely poor, even with projection of parameter estimates.

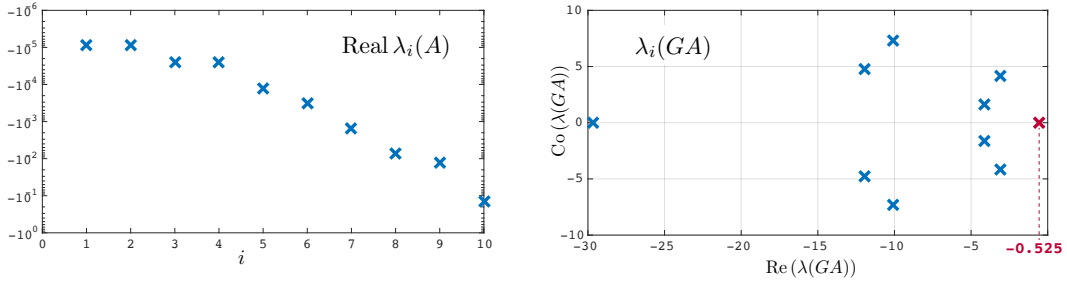

Figure 6: Eigenvalues of $A$ and $GA$ for the finance example

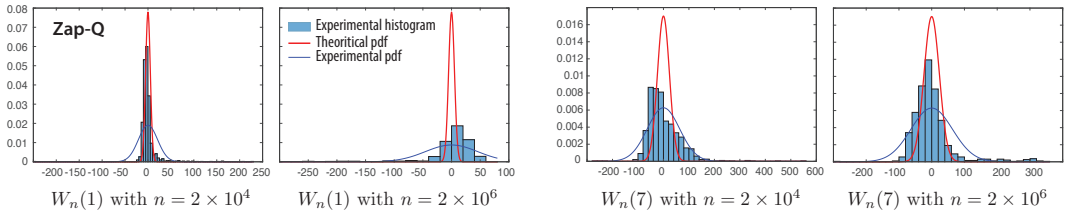

Figure 7: Theoretical and empirical variance for the finance example

In applying the Zap Q-learning algorithm it was found that the estimates $\{\widehat{A}_n\}$ defined in the above recursion are nearly singular. Despite the unfavorable setting for this approach, the performance of the algorithm was better than any alternative that was tested. Fig. 7 contains normalized histograms of $\{W_n^i(k) = \sqrt{n}(\theta_n^i(k) - \overline{\theta}_n(k)) : 1 \leq i \leq N\}$ for the Zap-Q algorithm, with $\gamma_n \equiv \alpha_n^{0.85} \equiv n^{-0.85}$. The variance for finite $n$ is close to the theoretical predictions based on the optimal asymptotic covariance. The histograms were generated for two values of $n$, and $k = 1, 7$. Of the $d = 10$ possibilities, the histogram for $k = 1$ had the worst match with theoretical predictions, and $k = 7$ was the closest. The histograms for the $G$-Q(0) algorithm contained in [8] showed extremely high variance, and the experimental results did not match theoretical predictions.

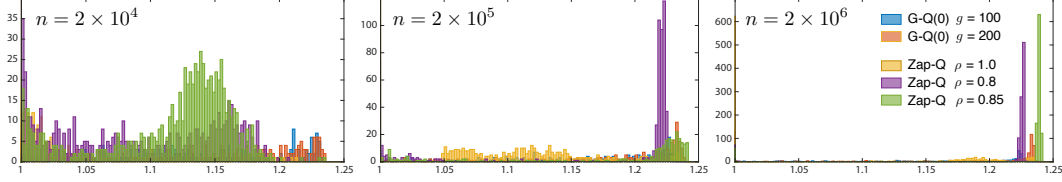

Figure 8: Histograms of average reward: $\boldsymbol{G}$-Q(0) learning and Zap-Q-learning, $\gamma_n \equiv \alpha_n^\rho \equiv n^{-\rho}$

| $n$ | 2e4 | 2e5 | 2e6 | 2e4 | 2e5 | 2e6 | 2e4 | 2e5 | 2e6 |
|---|---|---|---|---|---|---|---|---|---|
| $G$-Q(0) $g = 100$ | 82.7 | 77.5 | 68 | 81.1 | 75.5 | 65.4 | 54.5 | 49.7 | 39.5 |
| $G$-Q(0) $g = 200$ | 82.4 | 72.5 | 55.9 | 80.6 | 70.6 | 53.7 | 64.1 | 51.8 | 39 |
| Zap-Q $\rho = 1.0$ | 35.7 | 0 | 0 | 0.55 | 0 | 0 | 0 | 0 | 0 |
| Zap-Q $\rho = 0.8$ | 0.17 | 0.03 | 0 | 0 | 0 | 0 | 0 | 0 | 0 |
| Zap-Q $\rho = 0.85$ | 0.13 | 0.03 | 0 | 0 | 0 | 0 | 0 | 0 | 0 |

(a) Percentage of runs with $h^{\theta_n}(x) \leq 0.999$     (b) $h^{\theta_n}(x) \leq 0.95$     (c) $h^{\theta_n}(x) \leq 0.5$

Table 1: Percentage of outliers observed in $N = 1000$ runs. Each table represents the percentage of runs which resulted in an average reward below a certain value

Histograms of the average reward $h^{\theta_n}(x)$ obtained from $N = 1000$ simulations is contained in Fig. 8, for $n = 2 \times 10^4$, $2 \times 10^5$ and $2 \times 10^6$, and $x(i) = 1, 1 \leq i \leq 100$. Omitted in this figure are *outliers*: values of the reward in the interval $[0, 1)$. Table 1 lists the number of outliers for each run. The asymptotic covariance of the $\boldsymbol{G}$-Q(0) algorithm was not far from optimal (its trace is about 15 times larger than obtained using Zap Q-learning). However, it is observed that this algorithm suffers from much larger outliers.

## 4   Conclusions

Watkins' Q-learning algorithm is elegant, but subject to two common and valid complaints: it can be very slow to converge, and it is not obvious how to extend this approach to obtain a stable algorithm in non-trivial parameterized settings (i.e., without a look-up table representation for the Q-function). This paper addresses both concerns with the new Zap Q($\lambda$) algorithms that are motivated by asymptotic theory of stochastic approximation.

The potential complexity introduced by the matrix gain is not of great concern in many cases, because of the dramatic acceleration in the rate of convergence. Moreover, the main contribution of this paper is not a single algorithm but a class of algorithms, wherein the computational complexity can be dealt with separately. For example, in a parameterized setting, the basis functions can be intelligently pruned via random projection [2].

There are many avenues for future research. It would be valuable to find an alternative to Assumption Q3 that is readily verified. Based on the ODE analysis, it seems likely that the conclusions of Theorem 2.2 hold without this additional assumption. No theory has been presented here for non-ideal parameterized settings. It is conjectured that conditions for stability of Zap Q($\lambda$)-learning will hold under general conditions. Consistency is a more challenging problem.

In terms of algorithm design, it is remarkable to see how well the scalar-gain algorithms perform, provided projection is employed *and* the ratio of largest to smallest real parts of the eigenvalues of $A$ is not too large. It is possible to estimate the optimal scalar gain based on estimates of the matrix $A$ that is central to this paper. How to do so without introducing high complexity is an open question.

On the other hand, the performance of RPJ averaging is unpredictable. In many experiments it is found that the asymptotic covariance is a poor indicator of finite-$n$ performance. There are many suggestions in the literature for improving this technique. The results in this paper suggest new approaches that we hope will simultaneously

(i) Reduce complexity and potential numerical instability of matrix inversion,

(ii) Improve transient performance, and

(iii) Maintain optimality of the asymptotic covariance

**Acknowledgments:** This research was supported by the National Science Foundation under grants EPCN-1609131 and CPS-1259040.

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
