[Reviews · NeurIPS 2017]

Reviewer 1



The paper proposes a variant of Q-learning, called Zap Q-learning, that is more stable than its precursor. Specifically, the authors show that, in the tabular case, their method minimises the asymptotic covariance of the parameter vector by applying approximate second-order updates based on the stochastic Newton-Raphson method. The behaviour of the algorithm is analised for the particular case of a tabular representation and experiments are presented showing the empirical performance of the method in its most general form. This is an interesting paper that addresses a core issue in RL. I have some comments regarding both its content and its presentation. I will start with the latter. Given that the subject of the paper is dense, it is understandable that some effort is required to read it. However, even considering this factor, I think the paper could be reshaped a bit to become an easier, more accessible, read. I believe the opaqueness of the paper is due in part to three factors: - First, the emphasis seems to be too much on technicalities of the theoretical analysis rather than on the potential contribution to the field. For example, instead of motivating their method by first explaining the problem they are trying to solve in an easy, intuitive, way --which would reach a larger audience--, the authors resort to technical arguments right at the introduction (cf. paragraph starting in line 18). As an another example, item (iii) of Theorem 2.2, one of the main results of the paper, is never discussed in words. - Second, and more important, the structure and clarity of the text could be improved. As the paper is an excerpt of a longer article, some passages of the text feel a bit disconnected, excessively compressed or not very well-explained. For example, it is unclear to me what exactly the authors mean by “projections” in lines 161 and 163. - Finally, both the terminology and the notation adopted in the paper are a bit unusual in the RL literature. As for the content of the paper, I only have two general observations: - First, given the recent successful applications of RL using nonlinear function approximators, I wonder how difficult it would be to extend the proposed approach to that scenario. Algorithm 1 seems to have been derived considering a linear model, but if we make lambda = 0 it might not be too difficult to generalise the update equations to complex nonlinear models like neural networks. I think the authors should at least briefly discuss this subject in the paper. - I am also curious about how the proposed method compares against other algorithms that also use approximate second-order information in their update equations, like RMSProp or Adam --especially considering that they can easily be applied to more general, nonlinear, approximators.

Reviewer 2



The authors propose a new class of Q-Learning algorithms called Zap Q(\lambda) which use matrix-gain updates. Their motivation is to address the convergence issue with the original Watkins' Q Learning Algorithm. The algorithm involves having to perform matrix inversion for the update equations. The first set of experiments are demonstrated on a 6 node MDP graph showing Zap Q converging faster and also with low Bellman error. The second set of experiments is on a Finance Model with a 100 dimensional state space investigating the singularity of the matrix involved in the gain update equations. Though the ideas are interesting and research extending the fundamental algorithms are very necessary, I would have been happier to see more conventional RL experiments (even if on small dimensional state spaces) like 2D grid world tasks or puddle world or discrete CartPole control. The Finance model was new to me and it's hard to clearly appreciate the underlying utility. Secondly, it would have been better if the authors discussed how this class of algorithms can be used in parametrized high dimensional settings due to the computational cost of matrix inversion. An approximate version of that step is probably the bottleneck in this proposed class of algorithms becoming more commonly adopted for harder problems. I feel the paper would be more apt for a future conference if the authors could address these issues.

Reviewer 3



This paper proposed a new Zap Q-learning method, which is based on the newton-Raphson method for acceleration and variance reduction of stochastic approximation. Consistency, stability and asymptotical covariance analyses are provided. Experiments on small-scale MDP are also provided. The contribution seems to be the application of Newton-Raphson in the context of Q-learning. Several major comments: 1. In general, it is not clear how techniques from stochastic approximation can add to the literature of reinforcement learning. SA is concerned with stability and asymptotics, while RL is concerned with the efficiency of learning (sample complexity and regret). In general,I cannot convince myself why ppl care about the stability/asymptotic of SA in the context of RL. I think more justification is needed to bring together the theory of SA and RL. 2. The algorithm and the entire analysis relies on the key assumption that the underlying Markov decision process is "stationary," which essentially requires a stationary policy to be applied throughout. This is formally required as a condition Q1 in Theorem 1: (X,U) is an irreducible Markov chain. This assumption excludes the possibility of policy exploration and policy adaptation, which is key to RL. Under this theoretical limitation, the proposed method and analysis does not apply to general RL, making the stability/asymptotic results less interesting. 3. One contribution claimed in the paper is variance reduction. It seems that no theoretical justification is provided about how much is the variance reduced? Is it related to any condition number? Wha is the extra price paid by the new algorithm to achieve variance reduction? Minor comments: Line 14: emphasis in the literation is not the "computation" of finite-time PAC bound. The goal of RL is not to compute but to estimation and learn. This is a key difference between SA and RL. Line 26: what do you mean by the "gap"? Many notations and terminologies could be better explained, e.g., eligibility vectors, underline theta (underline Q is straightforward, but theta is not). Eq (5) is not necessary correct without appropriate assumptions on the parametric family. It doesn't even have to hold in general. Line 76: The operator mathcal(Q)^* is not necessarily a bijection unless some controbility/observability assumption is made? I am not 10% percent sure but it would be nice to be more precise.

Reviewer 4



This paper studies a second order methods for Q-learning, where a matrix used to determine step sizes is updated, and the parameters are updated according to the matrix. This paper provides conditions for the learning rate for the matrix and the learning rate for the parameters such that the asymptotic covariance of the parameters is minimized. Numerical experiments compare the proposed approach against Q-learning methods primarily without second order method. I have not checked the proofs in the supplementary material. The main contribution is in the proof that establishes the conditions for the optimal asymptotic covariance. I would like to see further discussion as to how this result relate to what is known in the prior work. For example, second order methods have been extensively studied with Q-learning or TD-learning, but how the asymptotic covariances of these method compare against the proposed approach, and how those existing methods differ from the proposed approach (e.g. with respect to the properties of gamma and alpha)? I see a comparison against [5] in L192-200 (which is out of place), but what about other existing methods? The baseline in numerical experiments is rather weak, and the advantages of the proposed approach against existing second order methods are unclear. Minor comments: L77: Provide a reference L78: Provide a reference L91: Define pmf L97: Provide representative references and discuss how the present paper relate to those. For example, Givchi and Phalhang, Quasi Newton Temporal Difference Learning, ACML 2014. Yao and Liu, Preconditioned Temporal Difference Learning, ICML 2008. Algorithm 1 Step 2: What is x, and how does it relate to X_n? Algorithm 1 Step 5: Define \gamma_{n+1} (12): What are the conditions for \alpha? L114: "more closely" than what? Figure 3: It is unclear what each panel shows? What is W_n(18) or W_n(10)? Figure 6: Horizontal axes should be better scaled for comparison. L225: What is "a stable algorithm"? L226: What are "non-trivial parameterized settings"? L236: What are "non-ideal parameterized settings"?

Reviewer 5



The paper makes the following contributions: 1) It shows that the asymptotic variance of tabular Q-learning decreases slower than the typical 1/n rate even when an exploring policy is used. 2) It suggests a new algorithm, Zap Q(lambda) learning to fix this problem. 3) It shows that in the tabular case the new algorithm can deliver optimal asymptotic rate and even optimal asymptotic variance (i.e., optimal constants). 4) The algorithm is empirically evaluated on both a simple problem and on a finance problem that was used in previous research. Significance, novelty --------------------- Q-learning is a popular algorithm, at least in the textbooks. It is an instance of the family of stochastic approximation algorithms which are (re)gaining popularity due to their lightweight per-update computational requirements. While (1) was in a way known (see the reference in the paper), the present paper complements and strengthens the previous work. The slow asymptotic convergence at minimum must be taken as a warning sign. The experiments show that the asymptotics in a way correctly predicts what happens in finite time, too. Fixing the slow convergence of Q-learning is an important problem. The fix, while relatively straightforward (but not entirely trivial), is important. My only worry is about the computational cost associated with matrix inversions. Even with incremental updates (Sherman-Morrison), the per-update cost is O(d^2) when we have d features. This may be OK in some applications, but it is a limitation (not a show-stopper by any means). The empirical evaluations are of sufficient depth and breadth for a conference publication proposing a new method (I actually like the way that they are conducted). The condition A3 seems strong (as noted by the authors themselves). I don't have high hopes for the generalization to the full linear feature-based case as in this Q-learning, as many of its relatives, tend not to have any strong guarantees (i.e., performance can be quite bad). However, this is not the problem of *this* paper. Soundness --------- I believe that the results are sound. I could not check the long version (yet) as I was asked to write a review just two days ago (as an extra review) and I did not have the time to check the proofs. From what I understand about these type of proofs, I believe that the proof can go through but there are some delicate steps (e.g., nonsmooth right-hand side of the ODE). That the authors discovered that A3 is necessary for their proof to go through at least shows that they are quite careful and attentive to details. (I am siding with them in that I believe it may be possible to remove A3, but this won't be trivial). The experiments are done with care; demonstrating nicely the effect of learning. Perhaps a sensitivity study could complement the experiments presented, but this is not critical in my opinion for a conference publication. Quality of writing ------------------ Very well written pose. Some may say that perhaps fewer symbols could be used, but at the end the notation was consistent and systematic and I did not find the paper too hard to read. Related work ------------ There is some similarity to [1-5], which would be good to cite. While these papers are definitely relevant (they seem to employ the same two-time scale logic in similar, TD/LMS context), the present paper is sufficiently different so that besides citing these as implementing the same idea in a different setting, no other modification is necessary to the paper. Minor comments -------------- * Perhaps Theorem 2.1 does not need Q2? * Line 120: "the gain sequence gamma" should be "the gain sequences gamma and alpha" * Line 124: The uninitiated reader may find it hard to interpret the phrase "An ODE approximation holds". * Numerical results: I expect the authors to share their code (github is excellent for this) or otherwise make it clear what parameter settings are used. A few details seem to be missing to replicate the experiments (e.g., what stepsize was used with RPJ?) * Line 151: "the Watkins'": remove "the" * Line 152: Which analysis does "this" refer to? (Middle of the page.) * Figure 3: I am guessing W_n is the error. I am not sure whether this was introduced beforehand. Maybe just remind the reader if it was. * Fig 4: Why is Speedy-Q not converging? Will it?? Overall vanilla Q-learning is not even that bad here.. * Line 174: "trails" should be "trials"? * Line 202: How is the condition number defined given that A is not even symmetric, so it may have complex eigenvalues? Magnitudes I assume? * Line 209: Hmm, I see W_n is defined here.. Move this up? References [1] H. Yao and Z-Q. Lie. Preconditioned temporal difference learning. In Proceedings of the 25th International Conference on Machine learning, 2008. [2] H. Yao, S. Bhatnagar, and Cs. Szepesvari. Temporal difference learning by direct preconditioning. In Multidisciplinary Symposium on Reinforcement Learning (MSRL), 2009. [3] Yao, H., Bhatnagar, S., and Szepesvári, Cs., LMS-2: Towards an Algorithm that is as Cheap as LMS and Almost as Efficient as RLS, CDC, pp. 1181--1188, 2009 [4] Pan, Yangchen, Adam M. White, and Martha White. "Accelerated Gradient Temporal Difference Learning." AAAI. 2017. [5] Givchi, A., & Palhang, M. (2014). Quasi Newton Temporal Difference Learning. ACML.